# Comparative Study of Algae-Based Measurements of the Toxicity of 14 Manufactured Nanomaterials

**DOI:** 10.3390/ijerph19105853

**Published:** 2022-05-11

**Authors:** Seung-Hun Lee, Kiyoon Jung, Jinwook Chung, Yong-Woo Lee

**Affiliations:** 1Department of Chemical and Molecular Engineering, Hanyang University, 55 Hanyangdaehakro, Ansan 15588, Korea; lck2394sin@hanyang.ac.kr (S.-H.L.); kyjung@hanyang.ac.kr (K.J.); 2R&D Center, Samsung Engineering Co., Ltd., Suwon 16523, Korea

**Keywords:** cell count method, chlorophyll fluorescence method, delayed fluorescence method, ecotoxicity, manufactured nanomaterial, *Raphidocelis subcapitata*

## Abstract

With the increasing use of nanomaterials in recent years, determining their comparative toxicities has become a subject of intense research interest. However, the variety of test methods available for each material makes it difficult to compare toxicities. Here, an accurate and reliable method is developed to evaluate the toxicity of manufactured nanomaterials, such as Al_2_O_3_, carbon black, single-walled carbon nanotubes (SWCNTs), multi-walled carbon nanotubes (MWCNTs), CeO_2_, dendrimers, fullerene, gold, iron, nanoclays, silver, SiO_2_, TiO_2_, and ZnO. A series of 72 h chronic and 8 h acute toxicity tests was performed using cell counting, chlorophyll, and delayed fluorescence methods. Comparable toxicities using the chlorophyll and delayed fluorescence methods were impossible to determine because the EC_50_ of some of the nanomaterials could not be measured. All three test methods were successfully applied to the chronic toxicity tests of manufactured nanomaterials, and cell counting was the only method applicable to acute toxicity tests. The toxicity data and the proposal of measurement method for manufactured nanomaterials obtained in this study can be helpful for preparing exposure standards and investigating the toxicities of other nanomaterials in the future.

## 1. Introduction

Manufactured nanomaterials are particulates with nanoscale (1 to 100 nm) measurements in at least one of the three physical dimensions or those intentionally manufactured with a specific surface area of at least 60 m^3^/cm^3^ [1]. Demand for such nanomaterials has been growing with the use of nanotechnology to manipulate materials at the atomic or molecular scale in the assembly of microscopic devices for cybernetics and medical applications. The market for manufactured nanomaterials is expected to reach USD 25 billion in the next five years [2]. However, threats posed by the unintended hazards of manufactured nanomaterials to human health and ecosystems have been increasing, giving rise to demands to regulate their use [3,4]. Because manufactured nanomaterials have very different properties from bulk-sized particles, the interest in manufactured nanomaterials is increasing. In particular, nanomaterials are known to be highly bioreactive due to their small size and large surface area. According to previous studies, when manufactured nanomaterials came into contact with a biological system, proteins, phospholipids, and DNA physically reacted, resulting in serious damage. In addition, carbon-based nanomaterials influenced the fibrous tissue and induced the generation of reactive oxygen species and the modification of protein by oxidative stress [5,6,7]. As the toxicity of manufactured nanomaterials was reported, the Sponsorship Program for the Testing of Manufactured Nanomaterials was launched by the Organisation for Economic Development and Co-operation (OECD) to ensure their safety for human use and the environment. The program selected 14 types of manufactured nanomaterials used primarily in industry and research—Al_2_O_3_, carbon black, single-walled carbon nanotubes (SWCNTs), multi-walled carbon nanotubes (MWCNTs), CeO_2_, dendrimers, fullerene, gold, iron, nanoclays, silver, SiO_2_, TiO_2_, and ZnO—for extensive and continued research on their level of toxicity [8]. Previous studies on the toxicity of manufactured nanomaterials used different test types, particle sizes, and methods of analysis, giving rise to difficulties comparing their degrees of toxicity. For example, Knauer et al. (2007) examined the ecotoxicity of carbon black using the alga types *Raphidocelis subcapitata*, but unlike other studies that tested the same material, Knauer et al. did not consider the particle size of the materials nor confirm their levels of toxicity [9]. Canesi et al. (2010), who conducted similar research on carbon black, fullerenes, SiO_2_, and TiO_2_ using the mussel *Mytilus galloprovincialis*, considered the particle size of the test materials and selected the point at which 50% of the cells turned red as a measure for toxicity, a criterion not applicable to other nanomaterials [10]. The variety of tested nanomaterial types and test methods makes it difficult to compare the findings of these two studies with those of the test types chosen by the OECD, such as algae and water fleas [9,10]. To compare toxicities among manufactured nanomaterials, a test method should be applicable to each test types. This study was designed to determine the most suitable test method among the three methods currently in use—cell counting, chlorophyll fluorescence, and delayed fluorescence—for the designated OECD test types of the green algae *R. subcapitata* in 72 h chronic toxicity and 8 h acute toxicity tests. In this study, toxicity measurements were achieved for all 14 types of manufactured nanomaterials included in the sponsorship program for testing manufactured nanomaterials, which had not been achieved before. These results can be helpful in preparing exposure and regulation standards and serve as sufficient reference data for the toxicity test of various manufactured nanomaterials.

## 2. Materials and Methods

### 2.1. Materials

To determine the toxicity of manufactured nanomaterials, we chose 14 types of manufactured nanomaterials among those proposed by OECD, although the sizes of these materials were not prescribed. The nanomaterials were chosen for this experiment due to their frequency of use in the industry and research [3,11,12,13,14,15,16,17,18]. The 14 types are Al_2_O_3_ (99%), CeO_2_ (>99.9%), gold (>99.9%), iron (>99%) silver (>99.9%), SiO_2_ (>99.5%), and ZnO (99%), all purchased from KoreaNano (Gwangmyeong, Korea), as well as MWCNTs (>98% carbon basis), dendrimers (G4-PAMAM dendrimer, 10wt.% in methanol), nanoclays (bentonite), and TiO2 (P25, >99.5%) purchased from Sigma-Aldrich (St. Louis, MO, USA). In addition, carbon black (>99%) was purchased from Uninanotech (Yongin, Korea), SWCNTs (>95%) from Carbon Nanotech (Pohang, Korea), and fullerene (C60, 99%) from Alfa Aesar (Haverhill, MA, USA). Additionally, physical and chemical properties of manufactured nanomaterials used in this study are shown in Table 1 [3,11,12,13,14,15,16,17,18].

### 2.2. Dispersion Treatment of Manufactured Nanomaterials

Several researchers reported that the toxicity mechanism was different depending on the particle size of the manufactured nanomaterial [19,20,21]. Accordingly, in order to compare the toxicity of manufactured nanomaterials under equal conditions, it was necessary to maintain and uniformly disperse the particle size of each material. For this purpose, 1500 mg/L of spray-dried gum arabic (Sigma Aldrich, St. Louis, MO, USA) was added to the culture medium as prescribed by OECD Test No. 201, to which the manufactured nanomaterials were added and sonicated [22]. Gum arabic is a dispersant mainly used for dispersing manufactured nanomaterials, and the dispersion was stably maintained without sediment even after several weeks had elapsed after addition [23,24,25,26]. A Power Sonic 510 sonicator bath (Hwashin Technology Company, Seoul, Korea) was used to disperse the manufactured nanomaterials at 40 kHz and at 20–25 °C. To maximize dispersion efficiency, the duration of sonication was varied by the material: 1 h for dendrimers, 2 h for SiO_2_, 24 h for SWCNTs and gold, and 4 h for other materials. The materials were then agitated at 200 rpm in a shaking incubator throughout the test period to maintain dispersion stability. The average dispersion and numerical size of particles were measured using a Litesizer 100 dynamic light-scattering apparatus (Anton Paar, Graz, Austria).

### 2.3. 72 h Chronic and 8 h Acute Toxicity Tests

The 72 h chronic and 8 h acute toxicity effects of each material on *R.*
*subcapitata* samples were conducted using OECD Test No. 201 (Freshwater Algae and Cyanobacteria, Growth Inhibition Test) [27]. All glassware and distilled water were sterilized at 120 °C for 15 min to minimize contamination. Three days before the test, the algae were inoculated to an initial concentration of 1 × 10^4^ cells/mL (72 h chronic test) and 20 × 10^4^ cells/mL (8 h acute test). Pre-cultured algae were used in the main test at the exponential growth stage. For the five treatment groups (excluding the control), the common ratio at each concentration was less than 3.2 as prescribed by OECD Test No. 201. In the case of the control group, after dissolving 1500 mg/L of gum arabic in the culture medium as in the treatment group, the algae were inoculated. The algae exposed to the test materials were placed in a shaking incubator, and the temperature and luminous intensity were maintained in accordance with culture conditions specified in OECD Test No. 201. The growth inhibition rate was measured every 24 h during 72 h of chronic testing and at hours 0, 1, 2, 4, and 8 during 8 h of acute testing using the cell counting, chlorophyll fluorescence, and delayed fluorescence methods. The results of the cell counting method, which used an NBS-80T optical microscope (Samwon, Yeongcheon, Korea), were tabulated by classifying the cell concentrations of the treatment and control groups according to the measurement time and the test material concentration. The growth inhibition rate (percentage inhibition in average specific growth rate) was determined by the ratio of the cell count of the control group to that of the treatment group. Results of the chlorophyll fluorescence tests were obtained using a chlorophyll fluorescence meter (TOXY-PAM, Walz, Effeltrich, Germany), and the chlorophyll fluorescence values of the control and treatment groups were classified by measuring the time and the concentration of the test material. Fluorescence inhibition rates were calculated using the difference between the maximum fluorescence amount (the maximum value of total fluorescence that can be emitted by chlorophyll pigments of algae, F_m_) generated in the treatment group with the maximum fluorescence amount of the control group [28]. Results of the delayed fluorescence method, which used a Type-6100 delayed fluorescence meter (Hamamatsu Photonics K.K., Hamamatsu, Japan), were obtained by classifying the delayed fluorescence values of the control and treatment groups according to measurement time and concentration of the test material. Fluorescence inhibition rates were determined by comparing the sum of all delayed fluorescence values from 1.1 to 60 s. The half-maximal effective concentration (EC_50_) was calculated to have a 95% confidence interval using the log-probit function in MedCalc (a toxicity calculation software package) based on experimentally derived growth inhibition rates.

## 3. Results and Discussion

### 3.1. Comparison of Manufactured Nanomaterials Using Dynamic Light Scattering

To verify the average distribution and numerical size of particles, dynamic light scattering was used to mark the EC_50_. The particle distribution and average particle size of each material are shown in Table 2. A comparison of average particle sizes before and after dispersion revealed a 21-fold increase after coagulation in every material, implying that gum arabic had negatively charged the particles in water [29]. Because all the manufactured nanomaterials have a negative charge in water, the gum arabic was wrapped around the agglomerated particles. To minimize the effect of gum arabic during the ecotoxicity tests, the nanomaterial was first dispersed in distilled water. After the gum arabic was completely dissolved, a culture solution prepared following OECD Test No. 201 was added to maximize dispersion efficiency.

### 3.2. Results of 72 h Chronic Toxicity Tests

In the 72 h chronic toxicity tests, EC_50_ measurements were obtained for all 14 types of manufactured nanomaterials (Figure 1). However, the toxicity of gold failed to reach EC_50_ even though the maximum concentration (200 mg/L) was stably dispersed for 72 h. Thus, the EC_50_ of gold nanoparticles was applied to the calculated value. During measurements of chlorophyll fluorescence, the unique fluorescence of manufactured nanomaterials was observed between wavelengths of 430 and 660 nm, which is the region within which chlorophyll becomes fluorescent [28,30]. To address this problem, a solution was prepared for each tested concentration of the nanomaterials without algae, and the measured fluorescence of the nanomaterials at each test time was subtracted from the fluorescence of the sample to calculate only the fluorescence of the algae. For the delayed fluorescence method, no fluorescence of nanomaterials was observed. The fluorescence measurement method applies to a range of concentrations that depend on the nanomaterial of each device. Nanomaterials tested at concentrations higher than this range were diluted to the measurable range. In the case of gold, because the EC_50_ could not be measured at the maximum dispersible concentration with the cell counting and chlorophyll fluorescence methods, toxicities were compared using calculated values. When the magnitude of toxicity was compared using the cell counting method, the results, from the highest to the lowest, were in the order of ZnO > silver > carbon black > MWCNTs >SWCNTs > dendrimers > CeO_2_ > Al_2_O_3_ > iron > SiO_2_ > TiO_2_ > gold (calculated) > fullerene > nanoclays. When chlorophyll fluorescence was used, the results in order of the magnitude of toxicity, from the highest to the lowest, were ZnO > silver > carbon black > MWCNTs > dendrimers > SWCNTs > CeO_2_ > Al_2_O_3_ > iron > SiO_2_ > TiO_2_ > gold (calculated) > fullerene > nanoclays. When the delayed fluorescence method was used, the results in order of toxicity, from the highest to the lowest, were ZnO > silver > carbon black > MWCNTs > dendrimers > CeO_2_ > SWCNTs > Al_2_O_3_ > iron > SiO_2_ > gold > TiO_2_ > fullerene > nanoclays (Figure 2). Griffitt et al. (2009) reported that toxicity was caused by chlorophyll fluorescence using *R. Subcapitata* for silver and TiO_2_ over 96 h. The EC_50_ of silver was 0.19 mg/L, which was similar to 0.3 mg/L measured in this experiment, but in the case of TiO_2_, the EC_50_ could not be detected. This was caused by the precipitation under an unstable dispersion during the test simply by sonication without using an appropriate dispersant [31]. On the other hand, in this study, the dispersion of TiO_2_ was stably maintained until the end of the experiment, and an EC_50_ measurement value of 218.4 mg/L was successfully obtained. In addition, the EC_50_ of SWCNTs and TiO_2_ using bovine serum albumin (BSA) as a dispersant were 30.0 mg/L for SWCNTs and 415 mg/L for TiO_2_, respectively [32,33]. These values were lower than 18.0 mg/L and 316.6 mg/L, as obtained from this study. This difference can be explained by the fact that manufactured nanomaterials adsorb with BSA to form BSA nanoparticles, and the toxicity level appears to be low [34]. This property of BSA was reported as a phenomenon that mitigated the toxicity level of various chemicals, including manufactured nanomaterials [35,36,37]. When EC_50_ measurements were examined, the toxicity of manufactured nanomaterials was relatively high in the three materials using the cell counting method, (carbon black, MWCNTs, and SiO_2_), in six materials using the chlorophyll fluorescence method (Al_2_O_3_, dendrimers, fullerene, iron, TiO_2_, and ZnO), and in five materials using the delayed fluorescence method (SWCNTs, CeO_2_, gold, nanoclays, and silver). Based on this, we supposed that there was no problem with applying all three test methods in the 72 h chronic toxicity test.

### 3.3. Results of 8 h Acute Toxicity Tests

Measurements of the EC_50_ for the 14 types of manufactured nanomaterials could be obtained from the 8 h acute toxicity test only by the cell counting method (Figure 3). As with the 72 h chronic toxicity test, unique fluorescence values for the nanomaterials were observed during chlorophyll fluorescence measurements. A solution was prepared for each test concentration of nanomaterials without algae, and the measured fluorescence of the nanomaterials at each test time was subtracted from the fluorescence of the sample to calculate the fluorescence of the algae. For the delayed fluorescence method, no fluorescence of nanomaterials was observed. Because the fluorescence measurement method involved a range of concentrations that could be measured depending on the nanomaterial for each device, nanomaterials tested at concentrations higher than this range were diluted to a measurable range. In acute toxicity tests, however, it is difficult to accurately measure florescence at a high test concentration because increasing the dilution factor can exceed the measurement limits of the instrument. With high dilution factors, the fluorescence inhibition rate could not be obtained, and the EC_50_ could not be calculated in three concentration ranges for Al_2_O_3_, iron, and TiO_2_. For carbon black, CeO_2_, SWCNTs, MWCNTs and nanoclays, the EC_50_ was calculated using only four values because the fluorescence inhibition rate within the error range could not be obtained for one concentration interval. In the case of gold, a calculated value was used to compare toxicities because the EC_50_ was not measured at the maximum dispersible concentration. When the magnitude of toxicity was compared using the cell counting method, the results in order of toxicity, from the highest to the lowest, were ZnO > silver > MWCNTs > carbon black > SWCNTs > dendrimers > iron > CeO_2_ > fullerene > TiO_2_ > SiO_2_ > gold (calculated) > Al_2_O_3_ > nanoclays. The magnitude of toxicity could not be compared using the chlorophyll fluorescence and delayed fluorescence methods, which were unable to produce EC_50_ values for Al_2_O_3_, iron, and TiO_2_. Therefore, we cannot describe the comparison of nanomaterials using the 8 h results.

These results were similar to those of previous studies [38], which concluded that the use of chlorophyll fluorescence measurements was inappropriate for manufacturing nanomaterials. This was due to the amount of fluorescence generated by the nanomaterial itself. In this case, a method of extracting and measuring chlorophyll was used to exclude the fluorescence of the manufactured nanomaterial, but chlorophyll was adsorbed to the nanomaterial. It is difficult to apply this approach to testing the toxicity of manufactured nanomaterials. With a delayed fluorescence, although self-fluorescence was not observed, the concentration of the manufactured nanomaterial was relatively high compared with the amount of delayed fluorescence, indicating a masking effect on the generated fluorescence. It is possible to reduce the occlusion effect by measuring the through-dilution of the sample, but it is impossible to measure some substances at high concentrations because the concentration of the algae also decreases. All 14 kinds of selected nanomaterials could be analyzed using the cell counting method. While a skilled analyst could be required, cell counting could be easily applied to the analysis of all manufactured nanomaterials, confirming cell counting as the optimal method for acute toxicity tests of manufactured nanomaterials.

## 4. Conclusions

Toxicity evaluation methods for manufactured nanomaterials (Al_2_O_3_, carbon black, single-walled carbon nanotubes (SWCNTs), multi-walled carbon nanotubes (MWCNTs), CeO_2_, dendrimers, fullerene, gold, iron, nanoclays, silver, SiO_2_, TiO_2_, and ZnO) were established using *R. subcapitata*, an official test types of the OECD. Cell counting, chlorophyll fluorescence, and delayed fluorescence methods were used, and 72 h chronic toxicity and 8 h acute toxicity tests were conducted using each method. The results strongly suggest that, while all three methods of measuring nanomaterial toxicity can be used in 72 h, the chronic toxicity tests of the 14 manufactured nanomaterials, only cell counting can be used in 8 h acute toxicity tests. Additionally, the toxicities of ZnO and silver were significantly higher than others. In this study, toxicity measurements were obtained for all 14 types of manufactured nanomaterials included in the OECD’s sponsorship program for testing manufactured nanomaterials, and this study focused on the toxicities of an individual manufactured nanomaterial. However, studies on the complex toxicity that actually affect the combination of various toxic substances are needed in the future.

## Figures and Tables

**Figure 1 ijerph-19-05853-f001:**
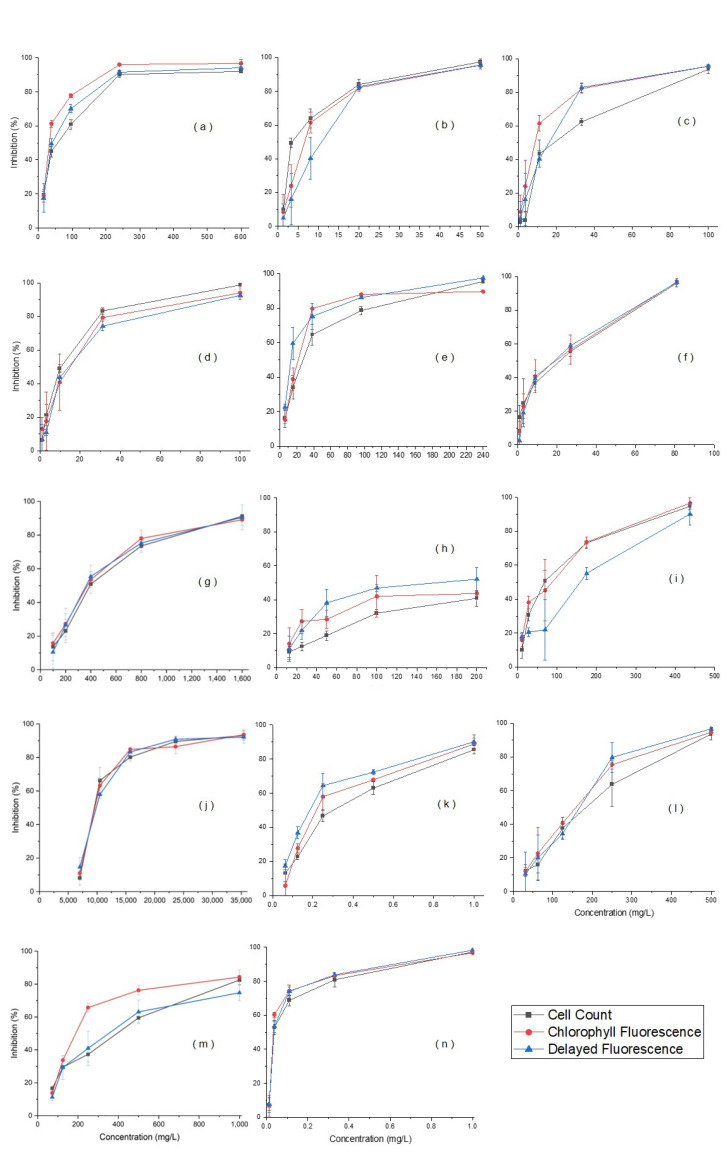
Inhibition of manufactured nanomaterials in 72 h chronic toxicity tests: (**a**) Al_2_O_3_; (**b**) carbon black; (**c**) SWCNTs; (**d**) MWCNTs; (**e**) CeO_2_; (**f**) dendrimers; (**g**) fullerene; (**h**) gold; (**i**) iron; (**j**) nanoclays; (**k**) silver; (**l**):SiO_2_; (**m**) TiO_2_; (**n**) ZnO.

**Figure 2 ijerph-19-05853-f002:**
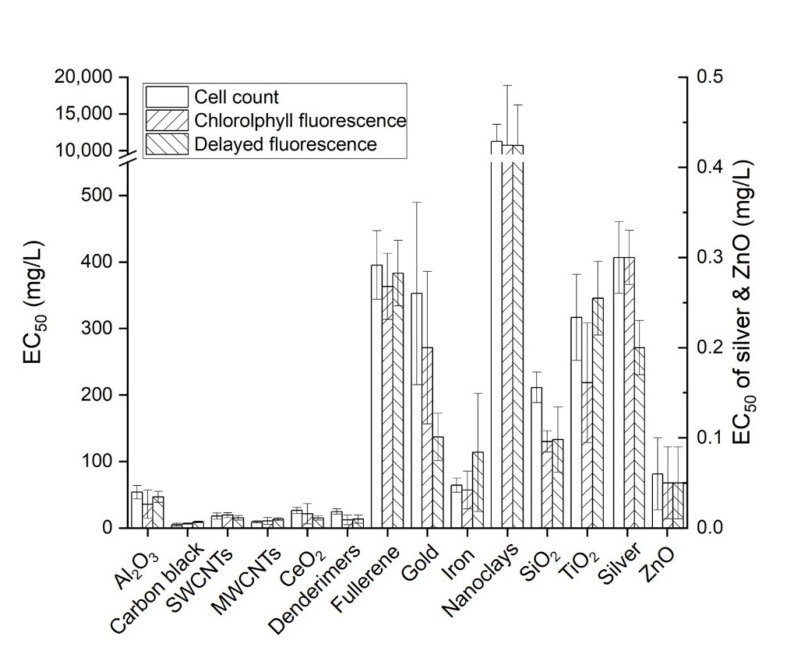
Comparison of EC_50_ values of manufactured nanomaterials.

**Figure 3 ijerph-19-05853-f003:**
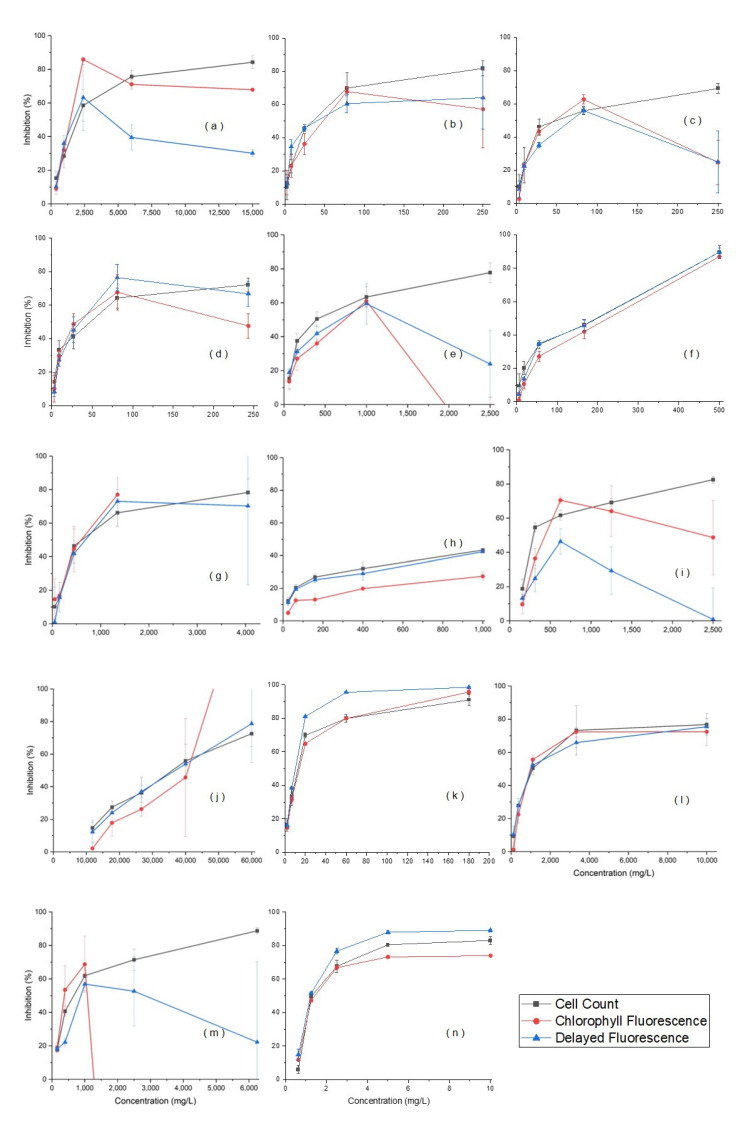
Inhibition of manufactured nanomaterials in 8 h acute toxicity tests: (**a**) Al_2_O_3_; (**b**) carbon black; (**c**) SWCNTs; (**d**) MWCNTs; (**e**) CeO_2_; (**f**) dendrimers; (**g**) fullerene; (**h**) gold; (**i**) iron; (**j**) nanoclays; (**k**) silver; (**l**) SiO_2_; (**m**) TiO_2_; (**n**) ZnO.

**Table 1 ijerph-19-05853-t001:** Physical and chemical characteristic of manufactured nanomaterials.

Manufactured Nanomaterials	Molecular Weight (g/mol)	Density (g/cm^3^)	Size	Solubility
Al_2_O_3_	101.96	3.987	20 nm	Insoluble
Carbon black	12.01	1.7	30 nm	Insoluble
SWCNT	N/A	1.3–1.4	D: 1–2 nm L: −10 μm	Insoluble
MWCNT	N/A	2.1	D: 5–50 L: 5–15 μm	Insoluble
CeO_2_	172.12	7.22	10–30 nm	Insoluble
Dendrimers	14,214.2	0.813	10 nm	Soluble
Fullerene	720.65	1.7–1.9	30 nm	Insoluble
Gold	196.97	19.3	15 nm	Insoluble
Iron	55.85	7.874	25 nm	Insoluble
Nanoclays	180.1	2.4	100 nm	Insoluble
Silver	107.87	10.49	20 nm	Insoluble
SiO_2_	60.084	2.1	15–20 nm	Insoluble
TiO_2_	79.87	4.23	21 nm	Insoluble
ZnO	81.38	5.61	35–45 nm	Insoluble

**Table 2 ijerph-19-05853-t002:** Change in average size of particles.

Manufactured Nanomaterials	Average Particle Size (nm)		Manufactured Nanomaterials	Average Particle Size (nm)	
	Before	After		Before	After
Al_2_O_3_	46.4 ± 2.3	2979.7 ± 29.8	Gold	30.4 ± 2.0	430.3 ± 11.1
Carbon black	93.1 ± 3.7	357.7 ± 56.7	Iron	30.4 ± 3.3	1664.7 ± 96.6
SWCNT	342.5 ± 34.3	973.8 ± 53.2	Nanoclays	149.8 ± 12.2	4058.0 ± 338.0
MWCNT	277.5 ± 30.5	1062.8 ± 78.8	Silver	30.2 ± 6.8	271.7 ± 18.3
CeO_2_	33.2 ± 0.3	212.9 ± 7.4	SiO_2_	30.5 ± 1.1	1719.8 ± 81.3
Dendrimers	8.8 ± 0.8	207.3 ± 6.0	TiO_2_	60.3 ± 0.8	300.1 ± 23.5
Fullerene	193.4 ± 16.4	593.1 ± 28.9	ZnO	31.8 ± 0.8	643.9 ± 86.8

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
