# Peer review of "Comparative Study of Algae-Based Measurements of the Toxicity of 14 Manufactured Nanomaterials"

_ijerph, 2022, doi:10.3390/ijerph19105853_

Round 1
Reviewer 1 Report
I suggest referring to several more relevant references in the introduction.
Author Response
I suggest referring to several more relevant references in the introduction.
We revised the introduction session and added relevant references:
“Because manufactured nanomaterials have very different properties from bulk-sized particles, the interest in manufactured nanomaterials is increasing. In particular, nanomaterials are known to be highly bioreactive due to their small size and large surface area. According to previous studies, when manufactured nanomaterials came into contact with a biological system, proteins, phospholipids, and DNA reacted physically and were resulted in serious damage. In addition, carbon-based nanomaterials influenced the fibrous tissue and induced the generation of reactive oxygen species and the modification of protein by the oxidative stress [5-7]. As the toxicity of manufactured nanomaterials was reported, the Sponsorship Program for the Testing of Manufactured Nanomaterials was launched by the Organisation for Economic Development and Co-operation (OECD) to ensure their safety for human use and to the environment.”

Reviewer 2 Report
Comparative Study of Algae-based Measurements of the Toxicity of 14 Manufactured Nanomaterials
This paper examines the 8- and 72hr toxicity of 14 nanomaterials to algae using OECD Test No. 21 and evaluates three methods of toxicity evaluation. This paper aims to establish which toxicity evaluation method is appropriate to create a standard for the evaluation of nanomaterial toxicity. The authors appear to have followed the OECD test accurately and have developed a dose-response relationship for growth inhibition for all 14 nanomaterials. Overall, the paper is well organized, and the methods of testing and analysis are appropriate. Given the sound scientific design to this experiment, the results are of scientific merit, and therefore I believe the study meets the expectations for publication in this journal. However, I do believe some representation of the data is missing. Therefore, I suggest the authors add a figure that clearly demonstrates the comparative toxicities of these materials, as stated in their objective and title of the study. Details on this matter, as well as some additional comments, mainly requesting more detail for their methods, are provided below.
Abstract
19: “half-maximal concentration’
I suggest including the term EC50, or median effects concentration. This is common terminology for toxicologists and regulators.
29: “60 m3/cm3”
Are these units correct?
Superscript ‘3’, i.e. m^3
Introduction
48: “Mytilus galloprovincialis” should be italicized if latin name
57: “R. subcapitata” same as above , i.e. italicize.
Materials and Methods
98: “specified culture conditions”
Provide details of these conditions or reference.
Tables 1 and 2: This information is of course relevant, but it’s represented in figures 1 and 2 since your data points align with the different treatment groups and their concentrations. So I suggest you place these tables in the SI. They are large and a bit of a distraction.
104: New paragraph, formatting error.
109-112: “Fluorescence inhibition rates were gauged by the maximum fluorescence (the maximum value of total fluorescence that algae chlorophyll pigments can emanate) in proportion to the cell concentration. “
This is still unclear to me how you calculated this. Did you normalize the fluorescence to the cell concentration, i..e. fluorescence / concentration (cells)?
General comment: Please provide detail on your control groups. Just a statement or two would suffice.
Results and Discussion
3.1 Dispersion treatment of manufactured nanomaterials.
I’m not clear on why the average particle size changes and why this is important. Perhaps it would help if you explained this in a bit more detail for those who are not familiar with handling nanomaterials. Do you want coagulation to occur with gum Arabic initially to create a more homogenous solution? And then before you add to the culture you dissolve the gum Arabic leaving just the nanomaterials?
146-147: “However, the EC50 for gold at the maximum 146 dispersible concentration was not actually measured, and a calculated value was used 147 instead. “
Explain why this was not actually measured. Why can’t it be measured?
157: “that”
Change to “than”
159-160 “toxicities were compared using calculated methods”
How were these toxicities calculated? Did you simply extrapolate based on the current trend of the dose-response curve?
175: “depending on the situation”
This is too vague. Need to be more specific. What does the situation have to be? What criteria need to be met to use these methods?
Figure 2: Labels (a ïƒ n) are missing.
Figures 1 and 2: I think some formatting can be done to improve these figures. Here are a few suggestions:
- Include only 1 legend for all graphs in each figure since they are the same
- Consider including y-axis title only on left hand side
- Consider including only x-axis at the bottom.
Add Figure 3: I suggest adding a third figure that specifically responds to your main objective here, i.e. to compare toxicities.
This figure could be a bar graph, EC50s on y-axis, nanomaterial names on x-axis. For each chemical you could have three bars (i.e. EC 50 for cell count, chlorophyll fluorescence, delayed fluorescence). I would focus on the 72-hr results, but you may wish to include a second figure for 8-hr results.
This way, you are presenting your findings and making it easy to compare the EC50 values for all nanomaterials. This is a better representation of your data.
Author Response
This paper examines the 8- and 72hr toxicity of 14 nanomaterials to algae using OECD Test No. 21 and evaluates three methods of toxicity evaluation. This paper aims to establish which toxicity evaluation method is appropriate to create a standard for the evaluation of nanomaterial toxicity. The authors appear to have followed the OECD test accurately and have developed a dose-response relationship for growth inhibition for all 14 nanomaterials. Overall, the paper is well organized, and the methods of testing and analysis are appropriate. Given the sound scientific design to this experiment, the results are of scientific merit, and therefore I believe the study meets the expectations for publication in this journal. However, I do believe some representation of the data is missing. Therefore, I suggest the authors add a figure that clearly demonstrates the comparative toxicities of these materials, as stated in their objective and title of the study. Details on this matter, as well as some additional comments, mainly requesting more detail for their methods, are provided below.
We revised and added the evidence of comparative toxicity to the text for the helping authors’ understanding.
- Abstract, line 19: “half-maximal concentration’
I suggest including the term EC50, or median effects concentration. This is common terminology for toxicologists and regulators.
Accordingly, we changed “half-maximal concentration” by “EC50”.
- Abstract, line 29: “60 m3/cm3”
Are these units correct?
Superscript ‘3’, i.e. m^3
We corrected the unit.
- Introduction, line 48: “Mytilus galloprovincialis” should be italicized if latin name
Throughout the whole manuscript, Mytilus galloprovincialis was italicized.
- Introduction, line 57: “R. subcapitata” same as above , i.e. italicize.
Throughout the whole manuscript, R. subcapitata was italicized.
- Materials and Methods, line 98: “specified culture conditions” Provide details of these conditions or reference.
According to referee’s comment, we changed “specified culture conditions” by “culture condition specified in OECD Test No. 201”
- Tables 1 and 2: This information is of course relevant, but it’s represented in figures 1 and 2 since your data points align with the different treatment groups and their concentrations. So I suggest you place these tables in the SI. They are large and a bit of a distraction.
According to referee’s comment, we deleted Tables 1 and 2. The sentence related to Tables 1 and 2 also was deleted.
- Materials and Methods, line 104: New paragraph, formatting error.
We corrected the mentioned formatting error.
- Materials and Methods, line 109-112: “Fluorescence inhibition rates were gauged by the maximum fluorescence (the maximum value of total fluorescence that algae chlorophyll pigments can emanate) in proportion to the cell concentration.” This is still unclear to me how you calculated this. Did you normalize the fluorescence to the cell concentration, i..e. fluorescence / concentration (cells)? General comment: Please provide detail on your control groups. Just a statement or two would suffice.
Accordingly, we added the description of control group to the text:
“in the case of the control group, after dissolving 1,500 mg/L of gum arabic in the culture medium as in the treatment group, the algae were inoculated.”
We demonstrated the calculation method for fluorescence inhibition rate:
“Fluorescence inhibition rates were calculated using the difference between the maximum fluorescence amount (the maximum value of total fluorescence that can be emitted by chlorophyll pigments of algae, Fm) generated in the treatment group with the maximum fluorescence amount of the control group [29].”
- Results and Discussion, 3.1 Dispersion treatment of manufactured nanomaterials.
I’m not clear on why the average particle size changes and why this is important. Perhaps it would help if you explained this in a bit more detail for those who are not familiar with handling nanomaterials. Do you want coagulation to occur with gum Arabic initially to create a more homogenous solution? And then before you add to the culture you dissolve the gum Arabic leaving just the nanomaterials?
We described the importance of average particle size in experiment as citing reference:
“Several researchers reported the toxicity mechanism was different depending on the particle size of the manufactured nanomaterial [20-22]. Accordingly, in order to compare the toxicity of manufactured nanomaterials under equal conditions, it was necessary to maintain and disperse the particle size of each material uniformly.”
We added the role of Arabic gum to the text:
“Gum arabic is a dispersant mainly used for dispersing manufactured nanomaterials, and the dispersion was maintained stably without sediment even after several weeks had elapsed after addition. [24-27].”
- Results and Discussion, line 146-147: “However, the EC50 for gold at the maximum 146 dispersible concentration was not actually measured, and a calculated value was used 147 instead.” Explain why this was not actually measured. Why can’t it be measured?
We revised the mentioned sentence:
“However, toxicity of gold failed to reach EC50 even though the maximum concentration (200 mg/L) that can be stably dispersed for 72 hours. Thus, the EC50 of gold nanoparticles was applied the calculated value.”
- Results and Discussion, line 157: “that” Change to “than”
We changed “that” by “than”
- Results and Discussion, line 159-160 “toxicities were compared using calculated methods”
How were these toxicities calculated? Did you simply extrapolate based on the current trend of the dose-response curve?
As long as we know, referee indicated the method for calculating toxicity. However, we already wrote Materials and Methods section as follows. In order to avoid redundant, we did not add the description of calculation methods.
“The half-maximal effective concentration (EC50) was calculated to have a 95% confidence interval using the log-probit function in MedCalc (a toxicity-calculation software package) based on experimentally derived growth inhibition rates.”
- Results and Discussion, line 175: “depending on the situation” This is too vague. Need to be more specific. What does the situation have to be? What criteria need to be met to use these methods?
We revised the mentioned sentence:
“Based on this, we suppose that there will be no problem in applying all three test methods in the 72-hour chronic toxicity test.”
- Results and Discussion, Figure 2: Labels (a ïƒ n) are missing.
We corrected label in Figure 2 and changed figure number by Figure 3.
- Results and Discussion, Figures 1 and 2: I think some formatting can be done to improve these figures. Here are a few suggestions:
Include only 1 legend for all graphs in each figure since they are the same
Consider including y-axis title only on left hand side
Consider including only x-axis at the bottom.
According to referee’s suggestion, we revised Figures 1 and 3.
- Results and Discussion, Add Figure 3: I suggest adding a third figure that specifically responds to your main objective here, i.e. to compare toxicities. This figure could be a bar graph, EC50s on y-axis, nanomaterial names on x-axis. For each chemical you could have three bars (i.e. EC 50 for cell count, chlorophyll fluorescence, delayed fluorescence). I would focus on the 72-hr results, but you may wish to include a second figure for 8-hr results. This way, you are presenting your findings and making it easy to compare the EC50 values for all nanomaterials. This is a better representation of your data.
According to referee’s suggestion, we added a new figure:
“Figure 2. Comparison of EC50 values of manufactured nanomaterials”
This figure includes only the results of the 72-hour test and also described the reason why the 8-hour test could not be compared:
“The magnitude of toxicity could not be compared using the chlorophyll fluorescence and delayed fluorescence methods, which were unable to produce EC50 values for Al2O3, iron, and TiO2. Therefore, we can not describe the comparison of nanomaterials using 8-hr results.”

Reviewer 3 Report
The paper is generally well-written with interesting results. A few comments for further improvements.
- Please provide more state-to-the-art literature in the introduction to highlight the toxicity impacts of nanomaterials. Examples: https://doi.org/10.3390/nano11123324 https://pubs.acs.org/doi/abs/10.1021/acsbiomaterials.8b00068
- Please highlight the research gaps and significance of the study more clearly in the introduction
- Please provide a summary table to highlight the major physical and chemical characteristics of 14 species of manufactured nanomaterials
- Please provide a more detailed discussion in the results section to highlight the result's similarities/discrepancies compared to previous studies.
- Please provide limitations of the study and recommendations for future studies in the conclusion section.
Author Response
The paper is generally well-written with interesting results. A few comments for further improvements.
- Please provide more state-to-the-art literature in the introduction to highlight the toxicity impacts of nanomaterials. Examples: https://doi.org/10.3390/nano11123324 https://pubs.acs.org/doi/abs/10.1021/acsbiomaterials.8b00068
We added the related references and revised the text:
“Because manufactured nanomaterials have very different properties from bulk-sized particles, the interest in manufactured nanomaterials is increasing. In particular, nanomaterials are known to be highly bioreactive due to their small size and large surface area. According to previous studies, when manufactured nanomaterials came into contact with a biological system, proteins, phospholipids, and DNA reacted physically and were resulted in serious damage. In addition, carbon-based nanomaterials influenced the fibrous tissue and induced the generation of reactive oxygen species and the modification of protein by the oxidative stress [5-7]. As the toxicity of manufactured nanomaterials was reported, the Sponsorship Program for the Testing of Manufactured Nanomaterials was launched by the Organisation for Economic Development and Co-operation (OECD) to ensure their safety for human use and to the environment.”
- Please highlight the research gaps and significance of the study more clearly in the introduction
We added the novelty and significance of this study to the text:
“In this study, toxicity measurements were achieved for all 14 types of manufactured nanomaterials included in the sponsorship program for testing manufactured nanomaterials, which had not been conducted before. These results can be helpful in preparing exposure and regulation standards and be a sufficient reference data for the toxicity test of various manufactured nanomaterials.”
- Please provide a summary table to highlight the major physical and chemical characteristics of 14 species of manufactured nanomaterials
According to referee’s comment, we added physical and chemical characteristic of manufactured nanomaterials as table:
“Table 1. Physical and chemical characteristic of manufactured nanomaterials.”
- Please provide a more detailed discussion in the results section to highlight the result's similarities/discrepancies compared to previous studies.
According to referee’s comment, we added a detailed discussion to the text:
“Griffitt et al. (2009) reported that toxicity results by chlorophyll fluorescence using R. Subcapitata for silver and TiO2 over 96 hours. EC50 of silver was 0.19 mg/L, which was similar to 0.3 mg/L measured in this experiment, but in the case of TiO2, the EC50 could not be detected. This resulted from the precipitation under unstable dispersion during test by simply sonication without using an appropriate dispersant [33]. On the other hand, in this study, the dispersion of TiO2 was maintained stably until the end of the experiment and an EC50 measurement value of 218.4 mg/L was successfully obtained. In addition, the EC50 of SWCNTs and TiO2 using bovine serum albumin (BSA) as a dispersant were 30.0 mg/L for SWCNTs and 415 mg/L for TiO2, respectively [34,35]. These values were lower than 18.0 mg/L and 316.6 mg/L obtained from this study. This difference can be explained that manufactured nanomaterials adsorb with BSA to form BSA-nanoparticles, and the toxicity level appears to be low [36]. This property of BSA has been reported that a phenomenon of mitigating the toxicity level of various chemicals including manufactured nanomaterials has been observed [37-39].
- Please provide limitations of the study and recommendations for future studies in the conclusion section.
According to referee’s comment, we added the proposal of further work to the text:
“In this study, toxicity measurements were obtained for all 14 types of manufactured nanomaterials included in the OECD's sponsorship program for testing manufactured nanomaterials, and this study was focused on toxicities of an individual manufactured nanomaterial. However, researches on the complex toxicity that actually affect the combination of various toxic substances are needed in the future.”

Reviewer 4 Report
I read carefully the article entitled Comparative Study of Algae-based Measurements of the Toxicity of Manufacured Nanomaterials submitted to IJERPH Journal. This manuscript is not well written and not clearly presented thus required substantial major revision to improve the quality of the manuscript.
- In Title manufacture spelling should be in correct form
- Abstract looks very general and not informative. In abstract authors should mention should mention the values of results and importance of research work in one or two sentences.
- Provide a nice graphical abstract representing the overview of the MS with key highlights.
- In the introduction section, write the novelty of the work and the problem statement clearly. More discussion about the selected NPs and also about the toxicity studies and their practical application is essential.
- In the whole manuscript unit style should be in standard format.
- Statistical analysis of the results should be provided in the materials and methods section.
- There is no standard deviation which makes question about the reproducibility of results.
- It’s important for all experimental work Report these values in the results and discussion
- Surprisingly there is no substantial discussion of results with the literature authors should concentrate on this issue during the revision stage.
- Techno Economic challenges and limitations of the developed system should be included? In addition, future research directions should include.
- The conclusion of the study needs to be added with the specific output obtained from the study, it could be modified with precise outcomes with a take home message.
- Some English and grammar mistakes are present that need to be correct to improve the quality of the manuscript.
Author Response
I read carefully the article entitled Comparative Study of Algae-based Measurements of the Toxicity of Manufacured Nanomaterials submitted to IJERPH Journal. This manuscript is not well written and not clearly presented thus required substantial major revision to improve the quality of the manuscript.
- In Title manufacture spelling should be in correct form
The expression “manufactured” has been confirmed an official name used since 2005 at the OECD official workshop (Report of the OECD Workshop on the Safety of Manufactured Nanomaterials Building Co-operation, Co-ordination and Communication). Therefore, we have decided to use the expression as it is.
- Abstract looks very general and not informative. In abstract authors should mention should mention the values of results and importance of research work in one or two sentences.
According to referee’s comment, we added the importance of this research to Abstract session:
“The toxicity data and the proposal of measurement method for manufactured nanomaterials obtained in this study can be helpful in preparing exposure standards and investigating toxicities of other nanomaterials in the future.”
- Provide a nice graphical abstract representing the overview of the MS with key highlights.
We added “graphical abstract”
- In the introduction section, write the novelty of the work and the problem statement clearly. More discussion about the selected NPs and also about the toxicity studies and their practical application is essential.
We added the novelty of this research to Introduction session:
“In this study, toxicity measurements were achieved for all 14 types of manufactured nanomaterials included in the sponsorship program for testing manufactured nanomaterials, which had not been conducted before. These results can be helpful in preparing exposure and regulation standards and be a sufficient reference data for the toxicity test of various manufactured nanomaterials.”
- In the whole manuscript unit style should be in standard format.
We revised unit style in standard format recommended from journal.
- Statistical analysis of the results should be provided in the materials and methods section.
We added the statistical analysis method used in this study to Material and methods session:
“The half-maximal effective concentration (EC50) was calculated to have a 95% confidence interval using the log-probit function in MedCalc (a toxicity-calculation software package) based on experimentally derived growth inhibition rates.”
- There is no standard deviation which makes question about the reproducibility of results. It’s important for all experimental work Report these values in the results and discussion
We added the average and standard deviation to Table 2.
- Surprisingly there is no substantial discussion of results with the literature authors should concentrate on this issue during the revision stage.
According to referee’s comment, we added a detailed discussion to the text:
“Griffitt et al. (2009) reported that toxicity results by chlorophyll fluorescence using R. Subcapitata for silver and TiO2 over 96 hours. EC50 of silver was 0.19 mg/L, which was similar to 0.3 mg/L measured in this experiment, but in the case of TiO2, the EC50 could not be detected. This resulted from the precipitation under unstable dispersion during test by simply sonication without using an appropriate dispersant [33]. On the other hand, in this study, the dispersion of TiO2 was maintained stably until the end of the experiment and an EC50 measurement value of 218.4 mg/L was successfully obtained. In addition, the EC50 of SWCNTs and TiO2 using bovine serum albumin (BSA) as a dispersant were 30.0 mg/L for SWCNTs and 415 mg/L for TiO2, respectively [34,35]. These values were lower than 18.0 mg/L and 316.6 mg/L obtained from this study. This difference can be explained that manufactured nanomaterials adsorb with BSA to form BSA-nanoparticles, and the toxicity level appears to be low [36]. This property of BSA has been reported that a phenomenon of mitigating the toxicity level of various chemicals including manufactured nanomaterials has been observed [37-39].
- Techno Economic challenges and limitations of the developed system should be included? In addition, future research directions should include.
This study is a comparative study on the toxicities and evaluation method of ecotoxicity for manufactured nano materials. Thus, it is difficult to present Techno Economic challenges and limitations of the developed system.
According to referee’s comment, we added the proposal of further work to the text:
“In this study, toxicity measurements were obtained for all 14 types of manufactured nanomaterials included in the OECD's sponsorship program for testing manufactured nanomaterials, and this study was focused on toxicities of an individual manufactured nanomaterial. However, researches on the complex toxicity that actually affect the combination of various toxic substances are needed in the future.”
- The conclusion of the study needs to be added with the specific output obtained from the study, it could be modified with precise outcomes with a take home message.
We revised conclusion session to avoid ambiguous wording and help readers understand.
“Toxicity evaluation methods for manufactured nanomaterials (Al2O3, carbon black, single-walled carbon nanotubes (SWCNTs), multi-walled carbon nanotubes (MWCNTs), CeO2, dendrimers, fullerene, gold, iron, nanoclays, silver, SiO2, TiO2, and ZnO) were established using R. subcapitata, an official test species of the OECD. The cell counting, chlorophyll fluorescence, and delayed fluorescence methods were used, and 72-hr chronic toxicity and 8-h acute toxicity tests were conducted using each method. The results strongly suggest that, while all three methods of measuring nanomaterial toxicity can be used in 72-h chronic toxicity tests of the 14 manufactured nanomaterials, only cell counting can be used in 8-h acute toxicity tests. Also, the toxicities of ZnO and silver were significantly higher than others. In this study, toxicity measurements were obtained for all 14 types of manufactured nanomaterials included in the OECD's sponsorship program for testing manufactured nanomaterials, and this study was focused on toxicities of an individual manufactured nanomaterial. However, researches on the complex toxicity that actually affect the combination of various toxic substances are needed in the future.”
- Some English and grammar mistakes are present that need to be correct to improve the quality of the manuscript.
As commented, we corrected those awkward grammars and improper wordings with the help of native speakers. In addition, we can provide the certification (from Eworld editing)

Round 2
Reviewer 3 Report
The manuscript has been revised accordingly.